# Chemical Screening of Metabolites Profile from Romanian *Tuber* spp.

**DOI:** 10.3390/plants10030540

**Published:** 2021-03-12

**Authors:** Adina-Elena Segneanu, Melinda Cepan, Adrian Bobica, Ionut Stanusoiu, Ioan Cosmin Dragomir, Andrei Parau, Ioan Grozescu

**Affiliations:** 1Department of Scientific Research and Academic Creation, West University of Timisoara, 300223 Timisoara, Romania; 2Cromatec-Plus, Scient Analytics, SCIENT, Research Center for Instrumental Analysis, 077167 Snagov, Romania; adrian.bobica@cromatec.ro (A.B.); ioangrozescu@gmail.com (I.G.); 3University Politehnica Timisoara, 300006 Timisoara, Romania; cepan.melinda@gmail.com (M.C.); istanusoiu@gmail.com (I.S.); 4Victor Babes University of Medicine and Pharmacy Timisoara, 300041 Timisoara, Romania; dr.dragomir.87@gmail.com (I.C.D.); parau.andrei@umft.ro (A.P.)

**Keywords:** secondary metabolites, truffles, GC-MS, mass-spectra, bioactive compounds

## Abstract

Truffles are the rarest species and appreciated species of edible fungi and are well-known for their distinctive aroma and high nutrient content. However, their chemical composition largely depends on the particularities of their grown environment. Recently, various studies investigate the phytoconstituents content of different species of truffles. However, this research is still very limited for Romanian truffles. This study reports the first complete metabolites profiles identification based on gas chromatography-mass spectrometry (GC-MS) and electrospray ionization quadrupole time-of-flight mass spectrometry (ESI-QTOF-MS) of two different types of Romania truffles: *Tuber magnatum pico* and *Tuber brumale*. In mass spectra (MS) in positive mode, over 100 metabolites were identified from 14 secondary metabolites categories: amino acids, terpenes, alkaloids, flavonoids, organic acids, fatty acids, phenolic acids, sulfur compounds, sterols, hydrocarbons, etc. Additionally, the biological activity of these secondary metabolite classes was discussed.

## 1. Introduction

At present, truffles (*Tuberaceae* family, *Tuber* genus) are considered an emblem of culinary refinement. Because of their nutritive and very particular organoleptic properties, they are considered as one of the most precious foodstuffs. Truffles were assigned mythical qualities in antiquity and then later in the Middle Age because they grow in the ground and are rarely found [1,2,3,4,5,6,7].

The high content of nutrients (proteins, fatty acids, minerals, amino acids) and, most of all, their recognizable flavor and aroma are, most probably, key factors that propelled these fungi into a highly precious and exclusive ingredient [1,2,3,4,5,6,7,8].

From ancient times, truffles have been considered aphrodisiacs. This property is attributed to the outstanding chemical constituents able to mime the male reproductive hormones (androsterone). There are reports about the truffle flavor is associated with perspiration, clay, garlic, mildew, and a faint onion smell [1,5]. There have been several studies on the volatile organic compounds (VOC) and the components involved in flavor. However, the chemical composition of truffles largely depends on the soil characteristics, environmental conditions, and especially the host trees [1,2,3,4,5,7,8,9,10,11,12].

The truffle’s growth in natural conditions depends on continuously changing climate conditions causing a restriction of their natural area, which directly influences their prices. Preserving truffles and their complex flavor still represents a challenge for the modern food industry, and is most probably the main factor in their market evaluation worldwide [1,3,8,9,10,11,12,13].

The increasing market demands (food, cosmetic industry) have brought forth new studies on the extension of truffle cultivation. The quality of truffles is attributed, in particular, to the different soil conditions (pH, organic substances, minerals, etc.), climate, and vegetation characteristics of each region [1,3,4,8,9,10,11,12,13]. It is appreciated that Central and South European forests have the highest phylogenetic variety, and are practically the origin growth area of these ectomycorrhizal fungi species. Romania is renowned in Europe for its truffle quality [13,14]. In Romania, the most widespread truffle variants are *Tuber brumale* and *Tuber aesetivum*. Nevertheless, in Romania, there are other types of truffles, such as *Tuber aestivum*, *Tuber Macrosporum*, *Tuber Mesentericum*, *Tuber magnatum pico,* and *Choiromyces meandriformis*. The more flavorful truffles (*Tuber magnatum pico* and *Tuber melanosporum*) are the most valuable. *Tuber magnatum pico* (white truffle), with a smooth garlic flavor, is considered one of the rarest varieties and cannot be cultivated. In South and Central Europe, *Tuber brumale* (winter truffle) can be found [13,14].

*Tuber* spp. are organisms adapted to habitats with a low concentration of oxygen by default. These symbiotic fungi most probably contain large quantities of antioxidant agents. The polyphenolics derivates from mushrooms induce a high antioxidant activity [3,15,16,17].

Recently, special attention was given to the potential biomedical application of hypogean fungus bioactive compounds, in particular, phytosterols, fatty acids, phenols, amino acids, volatile components, etc. [1,3,6,7,8,11,15,16,17,18,19,20,21,22,23,24,25,26,27,28,29]. However, still only a few scientific studies have been undertaken on secondary metabolites with therapeutic potential and on truffles’ biology [3,6,7,19,20,21,22,23,24,25,26,27,28,29].

There are relatively few studies on Romanian truffles, despite their high economic value being recognized. Furthermore, biologically active compounds from Romanian truffles have not been assessed through modern analytical methods. Research has only investigated the influence of soil particularities on truffle development [13]. Additionally, in a previous study, our team reported a comparative study on antioxidant activity through the electrochemical method (cyclic voltammetry), morphology (scanning electron microscopy), and semi-quantitative elemental analysis (EDAX) to estimate the diversity from two different types of truffles: *Tuber magnatum pico* and *Tuber melanosporum* [30].

The inclusion of the metabolomics approach in the study of secondary metabolites with therapeutic potential is paramount [31,32,33,34,35]. In this study, was used a qualitative untargeted metabolomics methodology based on the combination of gas-chromatography coupled with mass spectroscopy (GC-MS) and electrospray ionization quadrupole time-of-flight mass spectrometry (ESI-QTOF-MS) to analyze the metabolic profiles from two Romanian truffles species with high economic value, namely *Tuber magnatum pico* and *Tuber brumale*, or winter truffle.

## 2. Results and Discussion

The truffles chemical composition is highly complex and it is not yet fully described, especially since it is directly dependent on several factors, of which the most important are: host tree and soil parameters. Two solvents were selected with low polarity to achieve the extraction of truffles metabolites.

Thus, in dichloromethane, a polar aprotic solvent is expected to extract lipophilic compounds, such as fatty acids, terpenes, steroids, etc. Moreover, high polarity fractions (amino acids, alkaloids, carbohydrates, etc.) were extracted in methanol. The bioactive compounds screening from the truffles sample were tentatively identified by gas-chromatography coupled with mass spectroscopy (GC-MS) and electrospray ionization-quadrupole time-of-flight mass spectrometry (ESI-QTOF-MS) analysis.

Even though gas-chromatography coupled with mass spectroscopy (GC-MS) is one of the most common analytic techniques and is essential in the investigation of natural products due to their features, robustness and high sensitivity allow affordable and highly accurate separation and identification of metabolites [36].

Usually, gas-chromatography (GC) is used mainly for the separation of relatively low molecular weight metabolites such as amino acids, carbohydrates, organic acids, fatty acids, sterols, etc. [36].

A comparison of the total ion chromatographs of both truffle extracts presents the similarities and the differences regarding the metabolite types separated from the analyzed samples. The results are summarized in Table 1, which presents the GC-MS tentative compounds identification corresponding to *Tuber magnatum pico* and *Tuber brumale* samples.

### 2.1. Mass Spectrometry Analysis of Tuber magnatum pico and Tuber brumale

Truffle samples were diluted in methanol and characterized by ESI-TOF mass spectroscopy (ESI-QTOF-MS). The spectra revealed a complex mixture of molecules from which a few molecules were detected. Thus, mass spectra analysis showed the presence of 103 compounds in *Tuber magnatum pico* and 105 compounds from the *Tuber brumale*. Major of these phytochemicals are fatty acids, fatty esters, and sterols. The truffles samples were carried out in positive mode.

About 54% of the identified compounds were detected in the *m*/*z* range from 50 to 180. Identified compounds are listed in Table 2 and classified on the base of their *m*/*z* ratio (both theoretical and measured), chemical name, molecular formula, and the related literature. In sample 2 (*T. brumale*) another six additional compounds were detected: dipropyl trisulfide (*m*/*z*: 183.40), bis (2-methyl-3 furyl) disulfide (*m*/*z*: 227.34), sinapine (*m*/*z*: 311.37), ergosta-5,7,22-trien-ß-ol (*m*/*z*: 397.61), ergosta-5,7,22-trien-ß-ol (*m*/*z*: 397.66), and brassicasterol (*m*/*z*: 399.69).

The spectra disclose a very complex mixture of molecules from which only some molecules were detected. A total of 109 identified metabolites were attributed to different chemical classes such as amino acids, saccharides, flavonoids, aldehyde, ketone, esters, sulfur compounds, terpenoids, phenolic acids, steroids, hydrocarbons, and other data confirming results already published in the literature [7,10,15,17,19,20,21,22,23,24,25,26,27,28,29,35,36,37,38,39,40,41,42,43,44,45,46,47,48,49,50,51,52,53,54,55,56,57,58,59]. The results of the GC-MS were confirmed by ESI-QTOF-MS analysis.

The proportion of each metabolite categories distributed in two species truffles investigated was presented in the figures below. There is a distinction regarding the metabolite numbers accumulated in *T. brumale* (105), which was slightly larger than in *T. magnatum pico* (103). It was found that for *T. brumale*, the number of steroids and sulfur compounds was significantly higher than in *T. magnatum pico*. More amino acids were present in *T. magnatum pico* than *T. brumale*. In both truffle samples investigated, different amino acids were identified, and most of them are essential amino acids (valine, threonine, leucine, lysine, methionine) with few non-essential amino acids (ornithine, asparagine, cysteine) [7,25]. Previous studies revealed that each of these categories of metabolites identified in truffle samples exhibit biological activity [7,22,23,24,52]. For instance, sinapine, an alkaloid from *T. brumale*, possesses antioxidant and anti-inflammatory properties [7]. Aldehydes, alcohols, esters, and sulfur compounds are considered as responsible for the special truffle flavor [7,22,53,59]. Despite numerous studies, there is no complete description of the truffles’ very complex VOC assemble. Moreover, it is even more difficult to distinguish between each flavor component [7,10,38,40,45,53]. Some of them have been identified and presented in Table 3 [1,7,10,39,40,45]. In black truffles, such as *T. brumale*, the presence of sulfur compounds in large numbers is considered to be decisive for their specific aroma [1,7,10,39,40,45]. The environmental conditions lead to differences in the VOC profile between the same type of truffles harvested in different seasons.

Winter truffles have to develop more VOC molecules than white truffles, since the growing conditions are quite different between them [1,7,10,38,40,45]. Our results support this hypothesis. Among the winter truffles investigated, *T. brumale* contains more VOC molecules than white truffle, *T. magnatum pico*. Dipropyl trisulfide and bis (2-methyl-3 furyl) disulfide are the two sulfur compounds that have been identified only in our black truffle sample (*T. brumale*). More recently, truffles’ ergosteroid have been integrated into the VOC category with a characteristic sulfurous aroma [54]. Ergosta-5,7,22-trien-ß-ol, ergosterol, and brassicasterol were tentatively identified by ESI-QTOF-MS in *T. brumale*.

It should be mentioned that in both truffles, androstenone was identified, a steroidal pheromone with a distinct scent with various and completely different descriptions (floral, vanilla, sandalwood, sweaty, urine, or even without any odor [1,57]). It is estimated that due to the presence of this pheromone it is possible to train pigs or dogs to detect truffles [1,57], The predominant sulfur compounds in white truffle aroma are dimethyl sulfide and bis(methylthio)methane and dimethyl sulfide in black truffle aroma [40]. Disulfides derivates has bacteriostatic and antifungal properties [43]. The phenolic compound 4-aminophenol has shown to have an anti-inflammatory role [7].

Fatty acids were found in both truffles samples and represent a significant proportion of the total metabolites identified. Research has demonstrated that fatty acids have antibacterial and antimicrobial activity, as well as hypocholesterolemic properties [1,23,42,45]. Although absolute contents are, percentage-wise, basically the same (12%), the composition of terpenoids is varied and consists of squalene, β-elemene, α-terpineol, p-cymene, D-limonene, eucalyptol, thymol, lupenone, α-cubebene, 2-methyl-isoborneol, and lupeol. These compounds act mainly as antibacterial and antioxidant agents [7,45]. Moreover, previous investigations revealed that squalene present antibacterial, anticancer, antioxidant, tumoural protective, immunostimulant, and chemoprotective activity [23,45,46,47].

The steroid compounds found in truffles are involved in the mechanism of tumor protection and angiogenesis [7,23,26,46,47,48,49,50]. Furthermore, truffles contain stigmasterol and beta-sitosterol, compounds with similar chemical structures to cholesterol. Studies indicate that phytosterols act as hypercholesterolemic, immunomodulatory, and antitumor agents [52]. Recent studies report that ergosterol has shown antioxidant, anti-inflammatory, immunomodulating, and lowering hyperlipidemic effects [22,23,58,59].

The glycosylceramide identified in both truffles investigated is a sphingolipid type containing glucose residue [20,54]. This compound is highly bioactive with multiple roles in the organism: cell growth apoptosis, antitumor activity, and lowering cholesterol [20,54].

The flavor of the VOC metabolites identified in the investigated truffles is displayed in Table 3 and Figure 1. The key aroma of the investigated Romanian truffles is influenced by environmental conditions (soil parameters, tree host, etc.). Their fragrances are unique: medium sulfuric with sweet fruity, nutty, and floral notes [40].

### 2.2. Screening and Classification of Metabolites

A total of 109 metabolites were assigned to different chemical categories: amino acids, saccharides, nucleoside, flavonoids, organic acids, phenols and alcohol, esters, sulfur compounds, terpenoids and sesquiterpenes, aldehyde and ketones, phenolic acids, fatty acids, hydrocarbons, vitamins, alkaloids, and other (Table 4).

The data analysis reported in Table 4 allowed obtaining charts for *T. magnatum pico* and *T. brumale*, which are presented in Figure 2 and Figure 3.

## 3. Materials and Methods

Fresh fruiting bodies of *Tuber magnatum pico* (50 g) and *Tuber brumale* (50 g) were collected in late November 2019 from the area of the Eastern Carpathians and offered by Cromatec Plus after prior taxonomically and authentication. The truffles samples were rapid frozen in liquid nitrogen (−196 °C), ground and sieved to obtain a particle size lower than 0.5 mm, and kept at −80 °C to avoid enzymatic conversion or metabolites degradation.

For each analysis, 2 g of dried sample was subject to sonication extraction in 25 mL solvent (methanol/dichloromethane = 1:1) for 20 min at 45 °C, with a frequency of 50 kHz. The solution was concentrated using a rotavapor and the residue was dissolved in MeOH. The extract was centrifuged and the supernatant was filtered through a 0.2-μm syringe filter and stored at −18 °C until analysis.

### 3.1. Reagents

All used reagents were GC grade. Methanol and dichloromethane were purchased from VWR (Wien, Austria).

### 3.2. GC-MS Analysis

Gas chromatography was carried on the ClarusSQ8 GC/MS (PerkinElmer) apparatus with a nonpolar column Agilent 1909 s-433 (5% phenyl methyl siloxane); carrier gas, He, flow rate, 1 mL/min.

### 3.3. GC-MS Separation Conditions

The oven temperature program was 80 °C for 9 min, then raised to 220 °C (5 °C/min), to 280 °C (10 °C/min.), and finally held at this temperature for 20 min. The temperature of the injector was 260 °C and the temperature at the interface was 200 °C.

### 3.4. Mass Spectrometry

MS experiments were conducted on an EIS-QTOF-MS analysis from Bruker Daltonics, Billerica, MA, USA. All mass spectra were acquired in the positive ion mode within a mass range of (100–2500) *m/z*, with a scan speed of 2.1 scans/second. The source block temperature was kept at 80 °C. The reference provided in positive ion mode a spectrum with fair ionic coverage of the *m/z* range scanned in full-scan MS. The resulting spectrum is a sum of scans over the total ion current (TIC) acquired at 25–85 eV collision energy to provide the full set of diagnostic fragment ions.

Peak assignment to specific ion was based on the standard library, the NIST/NBS-3 (National Institute of Standards and Technology/National Bureau of Standards) spectral database. According to the peak, the resolution area was determined from the total ion current (TIC) or from the estimated selected ions integration. The results are presented in Table 1. The mass spectra of the compounds were compared with those from NIST/EPA/NIH Mass Spectral Library, and the identified compounds are presented in Table 2.

## 4. Conclusions

The proposed analytical methodology for the chemical screening of these Romanian truffles type allowed obtaining their metabolite profile. The number of metabolites (amino acids, steroids, and sulfur compounds) was different in both truffle species.

The different proportion of total metabolites identified between *T. brumale* and *T. magnatum pico* can be considered as evidence of the influence exerted by genetic and environmental conditions. Each of the chemical categories were detailed, including their biological activity. Moreover, we evaluated the profile of the key aroma compounds. However, studies on Romanian truffles are in the early stages considering that these fungi are still an unvalued source of compounds with high economic value. Further investigations are necessary to disclose the influence of the external factors (environmental condition, host tree, etc.) on the metabolic mechanism of truffles.

## Figures and Tables

**Figure 1 plants-10-00540-f001:**
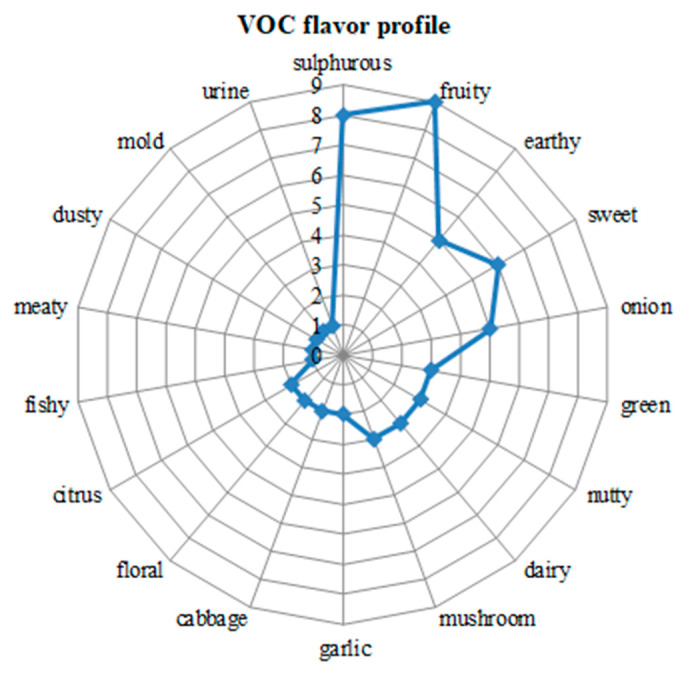
VOC flavor profile metabolites identified in truffle samples.

**Figure 2 plants-10-00540-f002:**
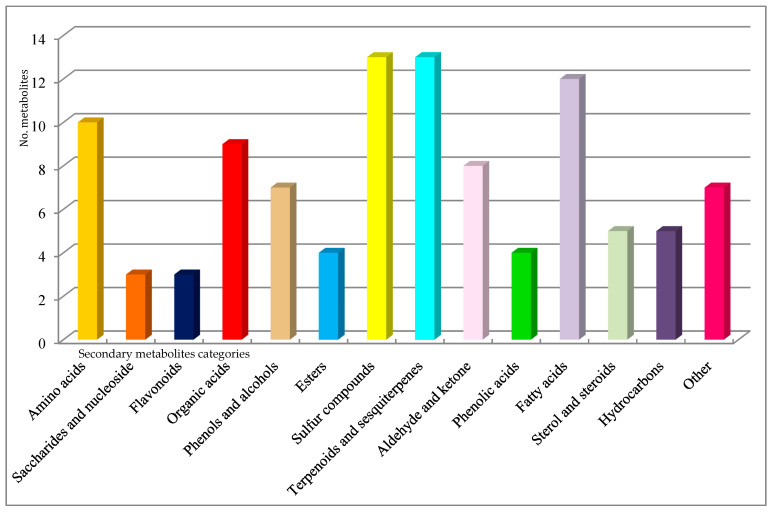
*Tuber magnatum pico*—metabolite classification.

**Figure 3 plants-10-00540-f003:**
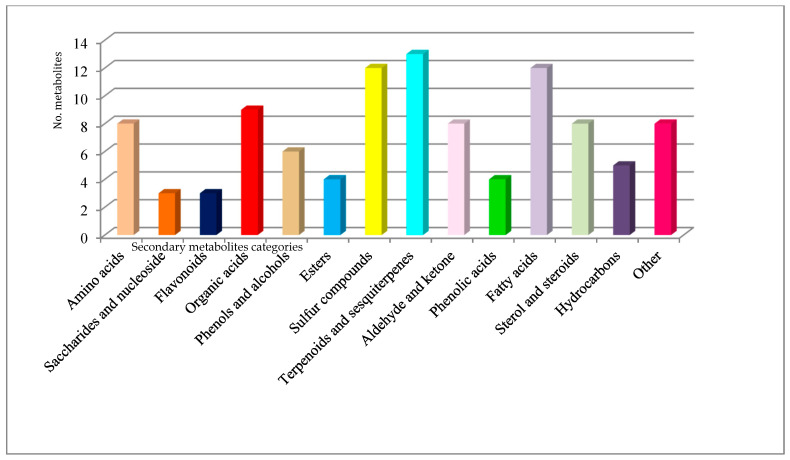
*Tuber brumale*—metabolite classification.

**Table 1 plants-10-00540-t001:** Main compounds identified by GC-MS analysis in both truffle samples.

Sample	Compounds Identified from GC-MS Library	RT	RI (Determinated)
*Tuber magnatum pico*	3-octanol	20.452	1087
	dimethyl sulfoxide	28.769	516
	stearic acid	32.974	216
	squalene	34.536	2745
	beta-sitosterol	36.167	3292
	campesterol	36.680	3297
	stearic acid	38.211	2163
	dimethyl sulfone	51.286	924
	benzothiazole	55.461	1184
*Tuber brumale*	3-octanol	20.452	1087
	1,2-butanediol	21.968	811
	lupeol	21.971	3265
	2,4-octanedione	35.445	1082
	tris(methylthio)methane	51.275	1364
	ergosterol	52.008	3085

**Table 2 plants-10-00540-t002:** Phytochemicals identified in both truffles sample detected by the MS method.

Sample Fraction	Compound No.	*m*/*z* Detected	Theoretic *m*/*z*	Formula	Tentative of Identification	Ref.
*Tuber magnatum pico*	1	65.17	65.15	C_2_H_6_S+	dimethyl sulfide	[3,7,10,23,37]
	2	89.18	89.15	C_6_H_12_O+	isoamyl alcohol	[7,10]
	3	89.14	89.12	C_4_H_8_O_2_+	3-hydroxy-2-butanone	[38]
	4	90.11	90.097	C_3_H_7_O_2_+	alanine	[7,25,39]
	5	95.15	95.14	C_2_H_6_O_2_S+	dimethyl sulfone	[10,37]
	6	95.23	95.20	C_2_H_6_S_2_+	dimethyl disulfide	[10,23]
	7	99.17	99.15	C_6_H_10_O	1-hexen-3-one	[38]
	8	105.19	105.18	C_4_H_8_OS	methional	[7,10,38]
	9	107.17	107.13	C_7_H_6_O+	benzaldehyde	[25,59]
	10	107.20	107.19	C_4_H_10_OS+	3-(methylthio)propanol	[38,40]
	11	109.07	109.06	C_7_H_8_O+	methoxybenzene (anisole)	[10]
	12	109.25	109.24	C_3_H_8_S_2_+	bis(methylthio)methane	[6,39]
	13	110.15	110.14	C_6_H_7_NO	4-amino-phenol	[7,10,22]
	14	117.17	117.16	C_6_H_12_O_2_+	butanoic acid ethyl ester	[39,40]
	15	117.19	117.17	C_6_H_12_O_2_	ethyl butyrate	[40]
	16	118.11	118.14	C_5_H_11_NO_2_+	valine	[7,25,39]
	17	120.08	120.03	C_4_H_9_NO_3_+	threonine	[7,10,25,39]
	18	121.18	121.16	C_8_H_8_O+	benzeneacetaldehyde	[38,59]
	19	123.07	123.67	C_8_H_10_O+	2-phenylethanol	[10]
	20	123.10	123.08	C_8_H_10_O+	p-cresyl methyl ether	[40]
	21	123.19	123.17	C_8_H_10_O	3-ethylphenol	[41]
	22	125.16	125.15	C_7_H_8_O_2_+	2-acetyl-5-methyl furan	[10,23,25]
	23	125.27	125.24	C_9_H_18_O+	nonanal	[10,59]
	24	127.16	127.13	C_8_H_14_O+	6-methyl-5-hepten-2-one	[38]
	25	127.23	127,21	C_8_H_14_O+	3,4-dimethyl-3-hexen-2-one	[38]
	26	127.29	127.27	C_2_H_6_S_3_+	dimethyl trisulfide	[3,38]
	27	129.21	129.18	C_10_H_8_+	naphthalene	[10]
	28	129.25	129.22	C_8_H_16_O+	1-octen-3-ol	[3]
	28	131.20	131.19	C_7_H_14_O_2_+	butanoic acid propyl ester	[38]
	29	132.17	132.75	C_5_H_12_N_2_O_2_+	ornithine	[7,10,25,38,39]
	30	132.19	132.18	C_6_H_13_NO_2_+	leucine	[7,10,25,38,39]
	31	133.08	133.06	C_4_H_8_O_3_+	asparagine	[7,10,25,38]
	32	135.25	135.23	C_10_H_14_+	p-cymene	[25,37,38]
	33	136.20	136.19	C_7_H_5_NS+	benzothiazole	[37,60]
	34	137.22	137.20	C_9_H_12_O	3-methyl-5-ethylphenol	[40,41,59]
	35	137.26	137.24	C_10_H_16_+	D-limonene	[38]
	36	137.27	137.25	C_10_H_14_+	cis-ocimene	[38]
	37	141.31	141.29	C_3_H_8_S_3_+	methyl(methylthio)dimethyl sulfoxide	[3,38]
	38	143.23	143.21	C_8_H_14_O_2_+	2,4-octanedione	[37]
	39	145.22	145.21	C_8_H_16_O_2_+	isobutyl hexanoate	[40]
	40	147.21	147.19	C_6_H_14_N_2_O_2_+	lysine	[7,10,25]
	41	149.19	149.17	C_9_H_8_O_2_+	cinnamic acid	[38]
	42	150.21	150.20	C_6_H_11_NO_2_S+	methionine	[7,10,25]
	43	151.23	151.22	C_10_H_14_O+	thymol	[21]
	44	155.27	155.25	C_10_H_18_O+	α-terpineol	[38]
	45	155.28	155.26	C_10_H_18_O+	eucalyptol	[38]
	46	155.35	155.32	C_4_H_10_S_3_+	tris(methylthio)methane	[3,41]
	47	156.18	156.16	C_6_H_9_N_3_O_2_+	histidine	[7,10,25,39]
	48	157.25	157.23	C_9_H_16_O_2_+	2-pentyl-3-butenoic acid	[59]
	49	159.26	159.25	C_9_H_18_O_2_+	2-isopropyl-hexanoic acid	[41]
	50	165.19	165.17	C_9_H_8_O_3_+	p-coumaric acid	[22]
	51	165.23	165.21	C_10_H_12_O_2_+	eugenol	[38]
	52	169.18	169.16	C_8_H_8_O_4_+	homogentisic acid	[22]
	53	169.31	169.29	C_11_H_20_O+	2-methylisoborneol	[21]
	54	171.15	171.13	C_7_H_6_O_5_+	gallic acid	[22,25]
	55	171.28	171.26	C_10_H_18_O_2_	3-methyl-2-nonenoic acid	[38,60]
	56	171.36	171.34	C_12_H_26_+	2,4-dimethyl-decane	[38]
	57	173.11	173.15	C_10_H_20_O_2_+	capric acid	[22,25,38]
	58	173.29	173.27	C_10_H_20_O_2_+	isobutyl hexanoate	[40]
	59	177.14	177.13	C_6_H_8_O_6_+	ascorbic acid	[22]
	60	179.28	179.24	C_11_H_14_O_2_+	benzene-1,2-dimethoxy-4-(2-propenyl)	[39]
	61	181.19	181.17	C_9_H_8_O_4_+	caffeic acid	[22,25]
	62	183.19	183.17	C_6_H_14_O_6_+	D-allitol	[51]
	63	187.24	187.22	C_12_H_10_O_2_+	2-naphthylacetic acid	[38]
	64	195.21	195.19	C_10_H_10_O_4_	ferulic acid	[7,10,25]
	65	205.36	205.35	C_15_H_24_+	α-cubebene	[10,38]
	66	205.37	205.35	C_15_H_24_+	caryophyllene	[10,39]
	67	205.38	205.36	C_15_H_24_+	β-elemene	[10,38]
	68	217.35	217.33	C_12_H_24_O_3_+	triisopropyl-S-trioxane	[3,38]
	69	227.36	227.35	C_14_H_26_O_2_+	8-dodecenyl acetate	[10,38]
	70	230.32	230.31	C_9_H_15_N_3_O_2_S+	L-ergothioneine	[7]
	71	235.40	235.39	C_15_H_26_N_2_+	sparteine	[7]
	72	239.35	239.34	C_16_H_18_N_2_	agroclavine	[7]
	73	241.33	241.31	C_6_H_12_N_2_O_4_S_2_+	cystine	[7,10,39]
	74	255.43	255.42	C_16_H_30_O_2_+	palmitoleic acid	[22,25]
	75	257.27	257.25	C_16_H_32_O_2_	palmitic acid	[22]
	76	273.45	272,43	C_19_H_28_O	androstenone	[52]
	77	278.25	278.24	C_9_H_17_NO_8_+	neuraminic acid	[7]
	78	281.41	281.40	C_18_H_32_O_2_	linoleic acid	[22,25]
	79	281.46	281.45	C_18_H_32_O_2_	octadecadienoic acid	[22,25,38]
	80	283.51	283.50	C_18_H_34_O_2_+	oleic acid	[22,25]
	81	289.47	289.45	C_18_H_36_O_2_+	stearic acid	[22,25]
	82	291.11	291.09	C_15_H_14_O_6_+	catechin	[21]
	83	298.30	298.28	C_11_H_15_N_5_O_5_+	7-methylguanosine	[7]
	84	300.27	300.29	C_18_H_37_NO_2_+	sphing-4-enine	[54]
	85	303.06	303.05	C_20_H_30_O_2_+	eicosapentaenoic acid	[22]
	86	305.53	305.51	C_20_H_32_O_2_+	arachidonic acid	[7,22]
	87	309.53	309.51	C_20_H_36_O_2_+	ethyl linolate	[21,22]
	88	322.38	322.36	C_11_H_19_N_3_O_6_S	S-methyl glutathione	[1]
	89	329.52	329.51	C_22_H_32_O_2_+	docosahexaenoic acid	[22]
	90	341.35	341.34	C_22_H_44_O_2_+	behenic acid	[22]
	91	343.32	343.31	C_12_H_22_O_11_+	trehalose	[22]
	92	369.62	369.61	C_24_H_48_O_2_+	lignoceric acid	[22,25]
	93	387.38	387.37	C_27_H_46_O+	cholesterol	[48,50,53,57,58,59,60]
	94	401.71	401.69	C_28_H_48_O	campestanol	[48,50,53,57,58,59,60]
	95	411.74	411.72	C_30_H_50_+	squalene	[7,23,45]
	96	413.71	413.70	C_29_H_48_O+	fucosterol	[48,50,53,57,58,59,60]
	97	415.73	415.71	C_29_H_50_O+	beta-sitosterol	[7,45]
	98	419.71	419.70	C_27_H_46_O_3_	cholest-5-en-3β,6,24S-triol	[48,50,53,57,58,59,60]
	99	425.72	425.70	C_30_H_48_O+	lupenone	[7,22,45]
	100	427.74	427.73	C_30_H_50_O	lupeol	[7,22,45]
	101	537.92	537.91	C_40_H_56_+	lycopene	[22]
	102	596.51	586.50	C_31_H_24_O_12_+	kolaflavanone	[7]
	103	812.72	812.70	C_46_H_89_NO_8_	glucosylceramide	[7,53,54]
*Tuber brumale*	1	95.23	95.20	C_2_H_6_S_2_+	dimethyl disulfide	[3,7,10,23,37]
	2	99.17	99.15	C_6_H_10_O	1-hexen-3-one	[38]
	3	105.19	105.18	C_4_H_8_OS	methional	[7,10,39]
	4	107.17	107.13	C_7_H_6_O+	benzaldehyde	[25,60]
	5	107.20	107.19	C_4_H_10_OS+	3-(methylthio)propanol	[38,40]
	6	109.07	109.06	C_7_H_8_O+	methoxybenzene (anisole)	[10]
	7	109.25	109.24	C_3_H_8_S_2_+	bis(methylthio)methane	[6,38]
	8	110.15	110.14	C_6_H_7_NO	4-amino-phenol	[7,10,22]
	9	117.17	117.16	C_6_H_12_O_2_+	butanoic acid ethyl ester	[38,41]
	10	117.19	117.17	C_6_H_12_O_2_	ethyl butyrate	[40]
	11	118.11	118.14	C_5_H_11_NO_2_+	valine	[10,25,39]
	12	120.14	120.13	C_4_H_9_NO_3_+	threonine	[7,10,25,39]
	13	121.18	121.16	C_8_H_8_O+	benzeneacetaldehyde	[38,59]
	14	123.07	123.67	C_8_H_10_O+	2-phenylethanol	[10]
	15	123.19	123.17	C_8_H_10_O+	3-ethylphenol	[41]
	16	123.10	123.08	C_8_H_10_O+	p-cresyl methyl ether	[40]
	17	125.16	125.15	C_7_H_8_O_2_+	2-acetyl-5-methylfuran	[10,23,25]
	18	125.27	125.24	C_9_H_18_O+	nonanal	[10,59]
	19	127.16	127.13	C_8_H_14_O+	6-methyl-5-hepten-2-one	[38]
	20	127.23	127.21	C_8_H_14_O+	3,4-dimethyl-3-hexen-2-one	[38]
	21	127.29	127.27	C_2_H_6_S_3_+	dimethyl trisulfide	[10,23]
	22	129.21	129.18	C_10_H_8_+	naphthalene	[10]
	23	129.25	129.22	C_8_H_16_O+	1-octen-3-ol	[3]
	24	131.20	131.19	C_7_H_14_O_2_+	butanoic acid propyl ester	[38]
	25	132.17	132.75	C_5_H_12_N_2_O_2_+	ornithine	[7,10,25,38,39]
	26	132.19	132.18	C_6_H_13_NO_2_+	leucine	[7,10,25,38,39]
	27	133.08	133.06	C_4_H_8_O_3_+	asparagine	[7,10,25,39]
	28	135.25	135.23	C_10_H_14_+	p-cymene	[25,37,38]
	29	137.22	137.20	C_9_H_12_O+	3-methyl-5-ethylphenol	[40,41,59]
	30	137.26	137.24	C_10_H_16_+	D-limonene	[38]
	31	137.27	137.25	C_10_H_14_+	cis-ocimene	[38]
	32	141.31	141.29	C_3_H_8_S_3_+	methyl(methylthio)dimethyl sulfoxide	[3,38]
	33	143.23	143.21	C_8_H_14_O_2_+	2,4-octanedione	[37]
	34	145.22	145.21	C_8_H_16_O_2_	isobutyl hexanoate	[40]
	35	147.21	147.19	C_6_H_14_N_2_O_2_+	lysine	[7,10,25]
	36	149.19	149.17	C_9_H_8_O_2_+	cinnamic acid	[39]
	37	150.21	150.21	C_6_H_11_NO_2_S+	methionine	[7,10,25]
	38	151.23	151.22	C_10_H_14_O+	thymol	[21]
	39	155.27	155.25	C_10_H_18_O+	α-terpineol	[38]
	40	155.28	155.26	C_10_H_18_O+	eucalyptol	[38]
	41	155.35	155.32	C_4_H_10_S_3_+	tris(methylthio)methane	[3,41]
	42	156.18	156.16	C_6_H_9_N_3_O_2_+	histidine	[7,10,25,39]
	43	157.25	157.23	C_9_H_16_O_2_+	2-pentyl-3-butenoic acid	[59]
	44	159.26	159.25	C_9_H_18_O_2_+	2-isopropyl-hexanoic acid	[41]
	45	162.15	162.13	C_7_H_15_NO_3_+	carnitine	[7]
	46	165.19	165.17	C_9_H_8_O_3_+	p-coumaric acid	[22]
	47	165.23	165.21	C_10_H_12_O_2_+	eugenol	[38]
	48	169.18	169.16	C_8_H_8_O_4_+	homogentisic acid	[22]
	49	169.31	169.29	C_11_H_20_O+	2-methylisoborneol	[21]
	50	171.15	171.13	C_7_H_6_O_5_+	gallic acid	[22,25]
	51	171.28	171.26	C_10_H_18_O_2_	3-methyl-2-nonenoic acid	[38,59]
	52	171.36	171.34	C_12_H_26_+	2,4-dimethyl-decane	[38]
	53	173.11	173.15	C_10_H_20_O_2_+	capric acid	[22,25,38]
	54	173.29	173.27	C_10_H_20_O_2_+	isobutyl hexanoate	[40]
	55	177.14	177.13	C_6_H_8_O_6_+	ascorbic acid	[22]
	56	179.28	179.24	C_11_H_14_O_2_+	benzene-1,2-dimethoxy-4-(2-propenyl)	[38]
	57	181.19	181.17	C_9_H_8_O_4_+	caffeic acid	[22,25]
	58	183.19	183.17	C_6_H_14_O_6_+	D-allitol	[51]
	59	183.40	183.38	C_6_H_14_S_3_+	dipropyl trisulfide	[10,23]
	60	187.24	187.22	C_12_H_10_O_2_+	2-naphthylacetic acid	[38]
	61	195.21	195.19	C_10_H_10_O_4_	ferulic acid	[7,10,25]
	62	205.36	205.35	C_15_H_24_+	α-cubebene	[10,38]
	63	205.37	205.35	C_15_H_24_+	caryophyllene	[7,38]
	64	205.38	205.36	C_15_H_24_+	β-elemene	[10,38]
	65	217.35	217.33	C_12_H_24_O_3_+	triisopropyl-S-trioxane	[3,38]
	66	227.34	227.30	C_10_H_10_O_2_S_2_	bis(2-methyl-3 furyl)disulfide	[40]
	67	227.36	227.35	C_14_H_26_O_2_+	8-dodecenyl acetate	[10,38]
	68	230.32	230.31	C_9_H_15_N_3_O_2_S+	L-ergothioneine	[7]
	69	235.40	235.39	C_15_H_26_N_2_+	sparteine	[7]
	70	239.35	239.34	C_16_H_18_N_2_	agroclavine	[7]
	71	241.03	241.31	C_6_H_12_N_2_O_4_S_2_+	cystine	[7,10,39]
	72	255.43	255.42	C_16_H_30_O_2_+	palmitoleic acid	[22,25]
	73	257.27	257.25	C_16_H_32_O_2_	palmitic acid	[22]
	74	273.45	272,43	C_19_H_28_O	androstenone	[53]
	75	278.25	278.24	C_9_H_17_NO_8_+	neuraminic acid	[7]
	76	281.41	281.40	C_18_H_32_O_2_	linoleic acid	[22,25]
	77	281.46	281.45	C_18_H_32_O_2_	octadecadienoic acid	[22,25,38]
	78	283.51	283.50	C_18_H_34_O_2_+	oleic acid	[22,25]
	79	289.47	289.45	C_18_H_36_O_2_+	stearic acid	[22,25]
	80	291.11	291.09	C_15_H_14_O_6_+	catechin	[21]
	81	298.30	298.28	C_11_H_15_N_5_O_5_+	7-methylguanosine	[7]
	82	300.27	300.29	C_18_H_37_NO_2_+	sphing-4-enine	[56]
	83	303.06	303.05	C_20_H_30_O_2_+	eicosapentaenoic acid	[22]
	84	305.53	305.51	C_20_H_32_O_2_+	arachidonic acid	[7,22]
	85	309.53	309.51	C_20_H_36_O_2_+	ethyl linolate	[21,22]
	86	311.37	311.36	C_16_H_24_NO_5_^+^	sinapine	[7]
	87	322.38	322.36	C_11_H_19_N_3_O_6_S	S-methyl glutathione	[1]
	88	329.52	329.51	C_22_H_32_O_2_+	docosahexaenoic acid	[22]
	89	341.35	341.34	C_22_H_44_O_2_+	behenic acid	[22]
	90	343.32	343.31	C_12_H_22_O_11_+	trehalose	[22]
	91	369.62	369.61	C_24_H_48_O_2_+	lignoceric acid	[22,25]
	92	387.38	387.37	C_27_H_46_O+	cholesterol	[48,50,53,57,58,59,60]
	93	397.61	397.60	C_28_H_44_O	ergosta-5,7,22-trien-ß-ol	[48,50,53,57,58,59,60]
	94	397.66	397.65	C_28_H_44_O	ergosterol	[51,53,57,58,59,60]
	95	399.69	399.67	C_28_H_46_O	brassicasterol	[7,45,48,50,53,57,58,59,60]
	96	401.71	401.69	C_28_H_48_O	campestanol	[7,45,48,50,53,57,58,59,60]
	97	411.74	411.72	C_30_H_50_+	squalene	[7,23,45]
	98	413.71	413.70	C_29_H_48_O+	fucosterol	[48,50,53,57,58,59,60]
	99	415.73	415.71	C_29_H_50_O+	beta-sitosterol	[7,45,48,50,53,57,58,59,60]
	100	419.71	419.70	C_27_H_46_O_3_	cholest-5-en-3β,6,24S-triol	[48,50,53,57,58,59,60]
	101	425.72	425.70	C_30_H_48_O+	lupenone	[7,22,45]
	102	427.74	427.73	C_30_H_50_O	lupeol	[7,45]
	103	537.92	537.91	C_40_H_56_+	lycopene	[22]
	104	596.51	586.50	C_31_H_24_O_12_+	kolaflavanone	[7]
	105	812.72	812.70	C_46_H_89_NO_8_	glucosylceramide	[1,7,54]

**Table 3 plants-10-00540-t003:** TOF-MS identified VOC odor compound in truffle samples.

No.	VOC Name	Odor
1	dimethylsulfone	sulfuric
2	dimethylsulfide	cabbage, sulfurous onion
3	dimethyl disulfide	cabbage, onion
4	methional	mold, French fry, yeasty
5	isoamyl alcohol	alcoholic, fruity
6	3-hydroxy-2-butanone	dairy, buttery
7	1-hexen-3-one	vegetable metallic
8	benzaldehyde	sweet almond
9	3-(methylthio)propanol	onion, garlic
10	methoxybenzene (anisole)	anise seed
11	bis(methylthio)methane	garlic sulfurous, mushroom
12	4-amino-phenol	sweet, balsamic
13	butanoic acid ethyl ester	sweet, fruity (apple)
14	ethyl butyrate	fruity, sweet
15	benzeneacetaldehyde	earthy, chocolate, floral
16	2-phenylethanol	floral
17	p-cresyl methyl ether	nutty, camphor
18	3-ethylphenol	phenolic
19	2-acetyl-5-methylfuran	nutty, dusty
20	nonanal	citrus
21	6-methyl-5-hepten-2-one	citrus, green, nutty
22	3,4-dimethyl-3-hexen-2-one	blue-cheese, nutty
23	dimethyl trisulfide	onion, leek
24	naphthalene	naphthalene
25	1-octen-3-ol	earthy, green, mushroom
26	butanoic acid propyl ester	fruity, pineapple
27	benzothiazole	sulfurous, nutty
28	3-methyl-5-ethylphenol	fruity
29	methyl(methylthio)dimethyl sulfoxide	sulfurous, broccoli
30	2,4-octanedione	earthy, dill
31	isobutyl hexanoate	sweet, fruity
32	tris(methylthio)methane	earthy, mushroom
33	carnitine	fishy
34	2-methylisoborneol	earthy, musty
35	3-methyl-2-nonenoic acid	fruity
36	isobutyl hexanoate	fruity, green
37	benzene-1,2-dimethoxy-4-(2-propenyl)	spicy, woody
38	dipropyl trisulfide	sulfurous, garlic, pungent
39	triisopropyl-S-trioxane	dairy
40	bis(2-methyl-3 furyl)disulfide	sulfurous, meaty
41	8-dodecenyl acetate	fruity, pineapple
42	androstenone	urine, sweet, floral
43	S-methyl glutathione	allium, sulfurous

**Table 4 plants-10-00540-t004:** Classification of metabolites identified in truffles samples on chemical categories.

Sample Fraction	Chemical Class	Metabolite Name
*Tuber magnatum* *pico*	Amino acids	alanine
		valine
		threonine
		ornithine
		leucine
		asparagine
		lysine
		methionine
		histidine
		cystine
	Saccharides and nucleoside	trehalose
		7-methylguanosine
		glucosylceramide
	Flavonoids	sparteine
		agroclavine
		kolaflavanone
	Organic acids	cinnamic acid
		2-pentyl-3-butenoic acid
		2-isopropyl-hexanoic acid
		p-coumaric acid
		3-methyl-2-nonenoic acid
		capric acid
		2-naphthylacetic acid
		neuraminic acid
		homogentisic acid
	Phenols and alcohols	4-amino-phenol
		isoamyl alcohol
		D-allitol
		2-phenylethanol
		3-ethylphenol
		1-octen-3-ol
		3-methyl-5-ethylphenol
	Esters	butanoic acid ethyl ester
		butanoic acid propyl ester
		ethyl butyrate
		8-dodecenyl acetate
	Sulfur compounds	dimethylsulfide
		dimethylsulfone
		dimethyl disulfide
		methional
		bis(methylthio)methane
		methyl(methylthio)dimethyl sulfoxide
		3-(methylthio)propanol
		tris(methylthio)methane
		triisopropyl-S-trioxane
		L-ergothioneine
		S-methyl glutathione
		dimethyl trisulfide
		benzothiazole
	Terpenoids and sesquiterpenes	p-cymene
		α-terpineol
		D-limonene
		cis-ocimene
		thymol
		eucalyptol
		2-methylisoborneol
		α-cubebene
		caryophyllene
		β-elemene
		squalene
		lupenone
		lupeol
	Aldehyde and ketone	benzaldehyde
		3-hydroxy-2-butanone
		benzeneacetaldehyde
		nonanal
		1-Hexen-3-one
		6-methyl-5-hepten-2-one
		3,4-dimethyl-3-hexen-2-one
		2,4-octanedione
	Phenolic acids	ferulic acid
		gallic acid
		caffeic acid
		catechin
	Fatty acids	palmitoleic acid
		palmitic acid
		linoleic acid
		octadecadienoic acid
		oleic acid
		stearic acid
		eicosapentaenoic acid
		arachidonic acid
		ethyl linolate
		docosahexaenoic acid
		behenic acid
		lignoceric acid
	Sterol and steroids	cholesterol
		campestanol
		fucosterol
		beta-sitosterol
		cholest-5-en-3β,6,24S-triol
	Hydrocarbons	2,4-dimethyl-decane
		2-acetyl-5-methylfuran
		naphthalene
		p-cymene
		eugenol
	Other	sphing-4-enine (ceramide)
		isobutyl hexanoate (fatty acid esters)
		ascorbic acid (vitamin)
		lycopene (carotenoid)
		benzene-1,2-dimethoxy-4-(2-propenyl)
		p-cresyl methyl ether
		methoxybenzene (anisole)
*Tuber brumale*	Amino acids	valine
		threonine
		ornithine
		leucine
		asparagine
		lysine
		methionine
		cystine
	Saccharides and nucleoside	trehalose
		7-methylguanosine
		glucosylceramide
	Flavonoids	sparteine
		agroclavine
		kolaflavanone
	Organic acids	cinnamic acid
		p-coumaric acid
		3-methyl-2-nonenoic acid
		capric acid
		2-naphthylacetic acid
		neuraminic acid
		homogentisic acid
		2-pentyl-3-butenoic acid
		2-isopropyl-hexanoic acid
	Phenols and alcohols	4-amino-phenol
		3-ethylphenol
		1-octen-3-ol
		3-methyl-5-ethylphenol
		2-phenylethanol
		D-allitol
	Esters	butanoic acid ethyl ester
		butanoic acid propyl ester
		ethyl butyrate
		8-dodecenyl acetate
	Sulfur compounds	dimethyl trisulfide
		benzothiazole
		methional
		bis(methylthio)methane
		methyl(methylthio)dimethyl sulfoxide
		3-(methylthio)propanol
		tris(methylthio)methane
		triisopropyl-S-trioxane
		L-ergothioneine
		S-methyl glutathione
		dipropyl trisulfide
		bis(2-methyl-3 furyl)disulfide
	Terpenoids and sesquiterpenes	p-cymene
		α-terpineol
		D-limonene
		cis-ocimene
		thymol
		eucalyptol
		2-methylisoborneol
		α-cubebene
		caryophyllene
		β-elemene
		squalene
		lupenone
		lupeol
	Aldehyde and ketone	benzaldehyde
		3-hydroxy-2-butanone
		benzeneacetaldehyde
		nonanal
		1-Hexen-3-one
		6-methyl-5-hepten-2-one
		3,4-dimethyl-3-hexen-2-one
		2,4-octanedione
	Phenolic acid	gallic acid
		ferulic acid
		caffeic acid
		catechin
	Hydrocarbons	2,4-dimethyl-decane
		2-acetyl-5-methylfuran
		naphthalene
		p-cymene
		eugenol
	Fatty acids	palmitoleic acid
		palmitic acid
		linoleic acid
		octadecadienoic acid
		oleic acid
		stearic acid
		eicosapentaenoic acid
		arachidonic acid
		ethyl linolate
		docosahexaenoic acid
		behenic acid
		lignoceric acid
	Sterol and steroids	cholesterol
		campestanol
		fucosterol
		beta-sitosterol
		cholest-5-en-3β,6,24S-triol
		ergosta-5,7,22-trien-ß-ol
		ergosterol
		brassicasterol
	Others	sphing-4-enine (ceramide)
		isobutyl hexanoate (fatty acid esters)
		ascorbic acid (vitamins)
		lycopene (carotenoid)
		benzene-1,2-dimethoxy-4-(2-propenyl)
		p-cresyl methyl ether
		Lycopene (carotenoid)
		Sinapine (alkaloid)

## Data Availability

All data are contained within the article.

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
