# Peer review of "Chemical Screening of Metabolites Profile from Romanian *Tuber* spp."

_plants, 2021, doi:10.3390/plants10030540_

Round 1

Reviewer 1 Report

The manuscript describes the metabolites profiling of two different species of truffles from Romania, Tuber magnatum pico and T. brumale.

The authors described the results of GC-MS and ESI-TOFMS analyses. The main comments are about the MS spectroscopic analysis.

In the figures 3 and 4 the authors showed two different m/z ranges of the two extracts, why? 

Moreover, they decided to analyse the metabolites composition through a direct infusion without a combination of liquid chromatography, an analytical approach widely applied. How they are able to identify more than 100 components in a complex mixture or noisy background (e.g. figure 4) without any separation, information about the retention time and polarity or comparison of retention time with standards? Furthermore, the authors mentioned in Materials and Methods section that they applied a collision energy ramp to get the full set of diagnostic fragment ions. How they attribute the fragmentation pattern to the single parent mass from the full scan MS? Please comment on that.

Author Response

Responses to the comments of Reviewer #1

Comment:

In the figures 3 and 4 the authors showed two different m/z ranges of the two extracts, why? 

Moreover, they decided to analyse the metabolites composition through a direct infusion without a combination of liquid chromatography, an analytical approach widely applied. How they are able to identify more than 100 components in a complex mixture or noisy background (e.g. figure 4) without any separation, information about the retention time and polarity or comparison of retention time with standards?  Furthermore, the authors mentioned in Materials and Methods section that they applied a collision energy ramp to get the full set of diagnostic fragment ions. How they attribute the fragmentation pattern to the single parent mass from the full scan MS? Please comment on that.

Authors’ Response

This observation is correct. We found your comments extremely helpful and have revised accordingly. However, the m/z ranges are similar for the both samples. The only difference that appears between the two samples refers to the extension of the range over 1400 m/z because values were detected after the value of 1200 m/z, an aspect that does not appear in the case of the first truffle analyzed.

According to source paper: Rapid MALDI-TOF MS identification of commercial truffles” (authors: Khalid El Karkouri, Carine Couderc, Philippe Decloquement, Annick Abeille, Didier Raoult and published in 2019 in Scientific Reports vol. 9, Article number: 17686) the MS can allow an rapid compunds identification from truffles.

However, we discuss this issue with experts in mass spectroscopy who also verified the results of this analysis. The conclusion was that it is not an error on compounds identification.

We mentioned at section Material and methods: „Peak assignment to specific ion was based on the standard library, NIST/NBS-3 (National Institute of Standards and Technology/National Bureau of Standards) spectral database. According to the peak, the resolution area was determined from total ion current (TIC) or from estimated from selected ions integration. The results are presented in Table 1.”

We apologize for our error and thank you for pointing out this problem. The mass spectra of compounds were compared with those from NIST/EPA/NIH EI-MS Library and the identified are presented in Table 2.

We have included this information in the revised version.

Coauthors and I very much appreciated the encouraging, critical and constructive comments on this manuscript by the reviewer. The comments have been very thorough and useful in improving the manuscript. Thank you!

Reviewer 2 Report

Very interresting and valuable topic of experiments. 

However manuscript requires several corrections. 

  1. The lyaout of the manuscript should be as follows: 1. Abstract 2. Introduction 3. Materials and Methods 4. Resluts or Results and Discussion 5. Referances. Materials and methodes MUST be before the section results because reader needs to know the research design before reads about the results and discussion.
  2.  Section Materials and Methodes needs to be rewritten. Missing the sampling and samples collection region. How many spcimens/ fruitng  were collected/ or how many grams of mushrooms were collected and how the sample was treated? Where it was collected? Methodes are mixed with results and discussion. Line 209-213 completely not understandable - use correct english.
  3. Abbreviations must be firstly used explained ie. VOC (?); ESI-QTOF (?) etc.
  4. Correct English languag ie. FUNGIES - plural from fungy or fungus is FUNGI or FUNGUSES. 
  5. Dot "." is misplaced in almoust every sentance when use quotations. 
  6. Use the same name or comon name or abreviation throighout the whole manuscript likr Tuber magnatum pico not only Tuber magnatum if refering to the same species. 
  7. The names of the tables and figures needs to be informative and explain themselves without the text. 
  8. Table 4 should be presented before figures 6 and 7
  9. Discussion is not exactly discussion more of the results presentattion mixed with methodes. Result section should explain the tables and figures and then this should be discussed - what is the outcome of this results. 
  10. I would not use word putative - because it means "Commonly belived or deemed to be the case; accepted by suppostition rather than as a result of proof" and you actually proved that those compounds exist in the fruiting bodies of two species of Tuber mushrooms. 

Author Response

Responses to the comments of Reviewer #2

Comment:

Very interresting and valuable topic of experiments. 

Authors’ Response: Thank you for your assessment.

Comment:

However manuscript requires several corrections. 

  1. The lyaout of the manuscript should be as follows: 1. Abstract 2. Introduction 3. Materials and Methods 4. Resluts or Results and Discussion 5. Referances. Materials and methodes MUST be before the section results because reader needs to know the research design before reads about the results and discussion.

Authors’ Response: We respectfully disagree with this comment. Our manuscript follows the order according to research manuscript sections from Plants Journal Instructions for Authors

Comment:

  1.  Section Materials and Methodes needs to be rewritten. Missing the sampling and samples collection region. How many spcimens/ fruitng  were collected/ or how many grams of mushrooms were collected and how the sample was treated? Where it was collected? Methodes are mixed with results and discussion. Line 209-213 completely not understandable - use correct english.

Authors’ Response: We apologize for our error and thank you for pointing out this problem. We have included information on the amount of truffles offered to the authors for the study  and samples collection region in the revised version.

However in manuscript at Section: Materials and methods we mention: “Fresh fruiting bodies of Tuber magnatum pico and Tuber brumale were collected in late November 2019 and offered by Cromatec Plus after prior taxonomically and  authentication”.  

Also, in the next paragraph, it is mentioned:  “The truffles samples were rapid freezing in liquid nitrogen (-196°C) ground and sieved to obtained particle size lower than 0.5 mm and kept at −80°C to avoid enzymatic conversion or metabolites degradation.

For each analysis, 2g of dried sample was subject to sonication extraction in 25 mL solvent (methanol/ dichloromethane =1:1) for 20 min. at 45°C with a frequency of 50 kHz. The solution was concentrated using a rotavapor and the residue was dissolved in MeOH. The extract was centrifuged and the supernatant was filtered through a 0.2-μm syringe filter and stored at −18°C until analysis

Comment:

  1. Abbreviations must be firstly used explained ie. VOC (?); ESI-QTOF (?) etc.

Authors’ Response: We thank the reviewer for pointing this out. We have revised accordingly.

Comment:

  1. Correct English languag ie. FUNGIES - plural from fungy or fungus is FUNGI or FUNGUSES. 

Authors’ Response: We thank the reviewer for pointing this out. We have revised accordingly.

Comment:

  1. Dot "." is misplaced in almoust every sentance when use quotations. 

Authors’ Response: We thank the reviewer for pointing this out. We have revised accordingly.

Comment:

  1. Use the same name or comon name or abreviation throighout the whole manuscript likr Tuber magnatum pico not only Tuber magnatum if refering to the same species. 

Authors’ Response: We thank the reviewer for pointing this out. We have revised accordingly.

Comment:

  1. The names of the tables and figures needs to be informative and explain themselves without the text. 

Authors’ Response: We thank the reviewer for pointing this out. We have revised accordingly.

Comment:

  1. Table 4 should be presented before figures 6 and 7

Authors’ Response: We thank the reviewer for pointing this out. We have revised accordingly.

Comment:

  1. Discussion is not exactly discussion more of the results presentattion mixed with methodes. Result section should explain the tables and figures and then this should be discussed - what is the outcome of this results. 

Authors’ Response: Thank you so much for your comments. The section title was restructured and hopefully is now clear for the reader.

The authors preferred, mainly due to the length of the article, to combine the Results section with Discussion (which follows the recommendations according to such as Research Manuscript Sections from Plants Instructions for Authors).

Comment:

  1. I would not use word putative - because it means "Commonly belived or deemed to be the case; accepted by suppostition rather than as a result of proof" and you actually proved that those compounds exist in the fruiting bodies of two species of Tuber mushrooms. 

Authors’ Response: We thank the reviewer for pointing this out. We have revised accordingly.

Coauthors and I very much appreciated the encouraging, critical and constructive comments on this manuscript by the reviewer. The comments have been very thorough and useful in improving the manuscript. Thank you!

We would like to thank the referee again for taking the time to review our manuscript.

Round 2

Reviewer 1 Report

The manuscript describing the metabolites profiling of two different species of truffles from Romania, Tuber magnatum pico and T. brumale has been revised based on the reviewer comments, which were properly addressed.

However, the main issue remains the MS analysis and compounds identification, in particular the library used for that.

The authors added the sentence “The mass spectra of compounds were compared with those from NIST/EPA/NIH EI-MS Library and the identified are presented in Table 2.” (lines 247-248). What seems unclear and inconsistent is that the authors performed an electrospray ionization- quadrupole time-of-flight mass spectrometry (ESI-QTOF-MS) analysis but they used for identification a database based on the electron ionization (EI), comparing two different ionization methods.

Please, the authors comment on that.

Author Response

Please see the attachment below. 

Round 3

Reviewer 1 Report

The present version of Segneanu et al. manuscript has been revised properly based on the previous comments.

Finally, I would only suggest to the authors to indicate in table 2 the adduct ion in the “m/z detected” column and remove the charge from the molecular formula of the identified compounds.

This manuscript is a resubmission of an earlier submission. The following is a list of the peer review reports and author responses from that submission.

Round 1

Reviewer 1 Report

"Chemical Screening of Metabolites Profile from Romanian Tuber spp" By Segneanuet et al. is interesting from the point of view of the instrumental methods used (TOF-MS), and this work could contribute for the characterization of Tuber brumale and Tuber magnatum. However, I personally think that the manuscript exists some major problem and innovative for the publication in Plants.

The manuscript lacks novelty, both instrumental methods have been previously implemented to T. brumale and T. magnatum, although the amount of data provided by the authors is important. There are many issues to be corrected or improved, but the biggest problem for me is the sampling method, just two grams of one truffle for each species, because of that I am really sorry, but I am forced to reject the article. You should include more samples, it is very easy to get them these days, truffles are not the “rarest species of edible fungi”.

For the future, if authors finally decide to analyze more samples, they should take into account the following aspects in this article:

Introduction

  • There are other studies on truffle metabolomics as El Karkouri et al. 2019 (https://doi.org/10.1038/s41598-019-54214-x), in which authors already analyzed brumale and T. magnatum. Or in lines 37 and 45, you should take in to account the maturity degree of truffles, more important than other factors in the chemical composition or metabolomics of truffles (Caboni et al. 2020. https://doi.org/10.1016/j.foodchem.2020.126573).
  • Please, include a citation in lines 49-52. And in these lines, please look at Genus/species, species always lower case and italics, and please, correct Tuber uncinatum according to index fungorum (It is a variety of aestivum).

Discussion

Authors mainly present some results, with just one sample and no more information about that sample it is impossible to assert anything. There is no deep discussion between the same or other truffles species that already published by other authors. Thus,

  • Please, Figure 1 and 2 with the same scale, and notice which peaks are representative of each species.
  • Line 154 and others, magnatum is commonly named “summer truffle”? Please look at Unece Truffle Standardshttps://unece.org/fileadmin/DAM/trade/agr/standard/standard/fresh/FFV-Std/English/53_Truffles.pdf.
  • Figures 6 and 7, please, name axis.
  • There are some grammatical mistakes as dots and commas, between others.

Material and Methods

  • Just one sample per truffle species is a very big problem, and thus, there is no information about them that can influence to the results obtained, such as Maturity or ripening stage. Some studies have detected changes in the biochemical characteristics of truffles associated to maturation (Harki et al., 2006, Caboni et al., 2020). Authors should establish the maturity degree of truffles according to Harki et al., 2006, and Zeppa et al., 2004.

Zeppa, S., Gioacchini, A. M., Guidi, C., Guescini, M., Pierleoni, R., Zambonelli, A., & Stocchi, V. (2004). Determination of specific volatile organic compounds synthesised during Tuber borchii fruit body development by solid-phase microextraction and gas chromatography/mass spectrometry. Rapid Communication in Mass Spectrometry, 18(2), 199-205.

Harki, E., Bouya, D., & Dargent, R. (2006). Maturation-associated alterations of the biochemical characteristics of the black truffle Tuber melanosporum Vitt. Food Chemistry, 99(2), 394–400.

Caboni, P., Scano, P., Sanchez, S., Garcia-barreda, S., Corrias, F., & Marco, P. (2020). Multi-platform metabolomic approach to discriminate ripening markers of black truffles (Tuber melanosporum). Food Chemistry, 319, 126573.

  • ºC is presented by 4 different ways at least. Please, be careful with little issues.

Conclusions

What is the conclusion? There is not a specific one.

Reviewer 2 Report

The manuscript “Chemical screening of metabolites profiles from Romanian tuber spp” is an interesting study on Romanian truffle, however it needs a carefully revision by a native English speaker/a professional language editing service to improve the grammar and readability.

  • Abstract Lines 12-13: Please change the sentence “but their chemical composition depends to a large extent on the particularities of their grown environment” with “but their chemical composition largely depend on the particularities of their grown environment”
  • Lines 37-38: please revise the sentence, is not clear. Why host trees is in brackets? The sentence doesn’t sound very good
  • Lines 75-76: please change the sentence “In this study, un qualitative untargeted metabolomics methodology” with “In this study, an untargeted qualitative metabolomics methodology”
  • In Figure 1 and Figure 2 is not clear if it is used the same scale of y axis. It would be better for a comparison, to report the figures using the same scale.
  • In the Figure 4 caption add the word “sample” after Tuber brumale
  • Line 145: check the word “ansamble”
  • Lines 145-149: please reformulate the sentence with a better English language
  • Line 208: “ground sieved” change with “ground AND sieved”
  • Line 212: “The solution was the concentration at rotavapor” change with “The solution was concentrated using a rotavapor”
  • A carefully revision of the English language by a native English speaker/a professional language editing service is necessary.

Reviewer 3 Report

Through GC-MS the authors study some secondary metabolites of two truffle species: Tuber magnatum pico and Tuber brumale.

Although the theme is interesting from a botanical point of view, due to the interaction of these species with multiple ecosystems and higher plants, the authors do not investigate these aspects. In fact, the manuscript is a discussion on the production of secondary metabolites without discussing whether this production may depend on particular ecological conditions. The authors should know that the production of secondary metabolites by the plant world is linked to “positive” or “negative” environmental stimuli. What are the stimuli that induce Tuber magnatum pico and Tuber brumale to produce the detected metabolites? Furthermore, can we also speak of VOC in the case of truffle species? What is the ecological significance?

Another aspect that shows a certain criticality is biological activity. In the abstract, the authors state "Also, the biological activity of these secondary metabolites classes was discussed." Unfortunately, no biological activity was measured. These aspects should be discussed by the authors.

I suggest to the authors some important references:

Kumar A et al. Plant behaviour: an evolutionary response to the
environment? Plant Biol (Stuttg). 2020 Nov;22(6):961-970. doi:
10.1111/plb.13149.

Quantitative and Qualitative Evaluation of Sorghum bicolor L. under
Intercropping with Legumes and Different Weed Control Methods,
https://doi.org/10.3390/horticulturae6040078

Germination and Seedling Growth Responses of Zygophyllum fabago, Salsola
kali L. and Atriplex canescens to PEG-Induced Drought Stress,
10.3390/environments7120107

In order to be able to publish this manuscript on PLANTS, authors should better discuss the issues raised above, otherwise their manuscript is more suitable in a food chemistry journal.